# Predictors of Step-Up Therapy and Outcomes in Idiopathic Granulomatous Mastitis: A Retrospective Cohort Study in Singapore

**DOI:** 10.3390/jcm14207157

**Published:** 2025-10-10

**Authors:** Kai Lin Lee, Jessele Shian Yi Lai, Peh Joo Ho, Hung Chew Wong, Karen Kaye Casida, Qin Xiang Ng, Mikael Hartman, Serene Si Ning Goh

**Affiliations:** 1NUS Yong Loo Lin School of Medicine, National University of Singapore, Singapore 117597, Singaporeserene_sn_goh@nuhs.edu.sg (S.S.N.G.); 2Saw Swee Hock School of Public Health, National University of Singapore and National University Health System, Singapore 117549, Singapore; 3Research Support Unit, Yong Loo Lin School of Medicine, National University of Singapore, Singapore 117549, Singapore; 4Department of General Surgery, Breast Division, National University Health System, Singapore 119228, Singapore; 5National University Cancer Institute of Singapore, National University Health System, Singapore 119074, Singapore

**Keywords:** idiopathic granulomatous mastitis, chronic, breast, inflammation, corticosteroids, methotrexate, predictive factors, inflammatory breast disease

## Abstract

**Simple Summary:**

Idiopathic granulomatous mastitis (IGM) is a rare, chronic inflammatory breast disease that often mimics malignancy and carries substantial morbidity. Corticosteroids are the usual first-line treatment, but many patients relapse or develop adverse effects, necessitating escalation to immunomodulatory therapy. In this context, methotrexate, an antimetabolite originally developed for oncology and widely used in autoimmune disorders, represents a promising drug repurposing candidate for IGM. In this study, we identified clinical predictors for requiring methotrexate and demonstrated that its use reduced relapse rates compared to corticosteroids alone. These findings highlight the potential of methotrexate as a therapeutic adjunct to improve outcomes in rare inflammatory breast diseases while minimizing cumulative steroid exposure.

**Abstract:**

**Background**: Idiopathic granulomatous mastitis (IGM) is a rare, benign, chronic inflammatory breast condition that poses diagnostic and therapeutic challenges. While corticosteroids are standard first-line therapy, some patients require additional immunomodulation, such as methotrexate. Predictive factors for step-up therapy remain poorly characterized. This study aimed to identify clinical, imaging, and pathological factors predictive of step-up therapy in IGM and evaluate associations between treatment approach and outcomes. **Methods**: A retrospective cohort study of women diagnosed with IGM was conducted between May 2022 and June 2024 at a tertiary center in Singapore. Data on demographics, clinical presentation, imaging, histopathology, and treatment were extracted. Step-up therapy was defined as methotrexate use following corticosteroids. Primary outcome was predictors of step-up therapy; secondary outcomes included treatment success, relapse, surgery, and time to remission. Statistical analyses included chi-square/Fisher’s exact tests, Cox models, and Kaplan-Meier analysis. **Results**: Fifty-two women (median age 39 years) were included; 26 (50%) required step-up therapy. Predictors included oral contraceptive (OCP) use (RR 1.92; 95% CI 1.45–2.53; *p* < 0.001), smoking (RR 2.00; 95% CI 1.49–2.69; *p* < 0.001), flares (RR 2.33; 95% CI 1.44–3.79; *p* = 0.002), and percutaneous aspiration (RR 2.10; 95% CI 1.53–2.88; *p* = 0.025). Patients receiving methotrexate had lower relapse rates (RR 1.23; 95% CI 1.12–1.36; *p* < 0.001) but longer time to remission (adjusted HR 0.09; 95% CI 0.02–0.46; *p* = 0.004). **Conclusions**: OCP use, smoking, flares, and aspiration need may predict step-up therapy in IGM. Early identification could guide a more personalized, potentially top-down treatment.

## 1. Introduction

Idiopathic granulomatous mastitis (IGM) is a rare, benign and chronic inflammatory breast disease primarily affecting women of child-bearing age, particularly those of Asian and Hispanic descent) [1,2,3,4,5]. First described by Kessler and Wolloch in 1972 [6], in clinical practice, IGM typically presents as a breast mass, sometimes accompanied by abscesses, sinus tracts, or erythema, and frequently mimics malignancy, which poses significant diagnostic and therapeutic challenges. IGM was detailed in a 5 case series by Cohen [7]. IGM has an annual prevalence of 2.4 in 100,000 women, an incidence rate of 0.37% [8,9], and comprises 24% of all inflammatory breast disease [10]. It is commonly treated with antibiotics, corticosteroids, immunosuppressants, such as methotrexate, and surgical excision [3,11,12,13,14,15]. Patients with IGM often experience a negative impact on their quality of life as this condition is known to recur frequently, run a prolonged disease course and is associated with breast pain [11].

The pathogenesis of IGM remains poorly understood [15]. While some cases are associated with infectious agents such as *Corynebacterium kroppenstedtii* [16,17], the prevailing hypothesis leans towards an autoimmune mechanism, supported by its response to corticosteroids and immunosuppressants. Standard first-line treatment typically involves corticosteroids, which can induce remission but are often associated with substantial side effects, including weight gain, mood changes, and glucose intolerance. Furthermore, some patients experience disease recurrence or fail to respond adequately to steroids alone [5]. Relapse rates after steroid therapy can be substantial, with reported rates ranging from 5% to over 50% [18,19,20].

Treatment of IGM is challenging due to the absence of standardized guidelines and the variability in disease course [1]. In this context, drug repurposing offers a rational strategy. Methotrexate, originally developed as an antineoplastic agent but now widely repurposed for autoimmune conditions such as rheumatoid arthritis, has shown promise as a steroid-sparing agent or adjunct in IGM. Its mechanism of action—folate antagonism with downstream effects on T-cell activity and pro-inflammatory cytokine production—provides a plausible molecular rationale for use in IGM, which is hypothesized to be an autoimmune-mediated disorder [21]. Current clinical practice often adopts a step-up approach, where methotrexate may be added to corticosteroids if symptoms persist or relapse occurs. However, despite some evidence that methotrexate is effective in preventing relapse and inducing remission of disease, methotrexate is rarely used in the first-line treatment of IGM [22]. Questions also remain regarding the potential role of methotrexate in the first-line treatment of IGM, either as monotherapy or in combination with corticosteroids.

Given the relapsing nature of IGM and the potential morbidity from prolonged corticosteroid use, there is a pressing need for further evidence and to identify predictive factors for treatment escalation. Early recognition of patients at risk for treatment failure could facilitate a more personalized, top-down approach, potentially improving outcomes and minimizing adverse effects. This study therefore aimed to describe the clinical, imaging, and pathological features associated with the need for step-up therapy and to examine the relationship between treatment modality and patient outcomes, and the findings may further inform IGM management strategies.

## 2. Materials and Methods

### 2.1. Study Design and Setting

This exploratory, retrospective cohort study was conducted at the National University Hospital (NUH), a tertiary referral center in Singapore, to evaluate predictive factors for step-up therapy and treatment outcomes in patients with IGM. The final cohort of 52 patients represents all eligible cases diagnosed and treated at NUH. The study period, in which patients were diagnosed and treated, spanned from May 2022 to June 2024. This study adhered to the Strengthening the Reporting of Observational Studies in Epidemiology (STROBE) guidelines [23].

### 2.2. Study Population

Inclusion criteria for the study were women aged 18 years and above, newly diagnosed, treatment-naïve and histopathologically confirmed IGM, and with at least 3 months of follow-up. The diagnosis required clinical evaluation, characteristic imaging features (e.g., irregular hypoechoic masses on ultrasound), and histopathological evidence of non-caseating granulomatous inflammation. Exclusion criteria were granulomatous mastitis of known cause, such as tuberculosis (ruled out with Ziehl-Neelsen staining and/or microbiological testing), fungal infections (excluded with periodic acid–Schiff and Grocott’s methenamine silver staining), recurrent IGM, foreign body reactions, incomplete records, or insufficient follow-up. Follow-up time is measured from the time of diagnosis.

### 2.3. Data Collection

Data were extracted from existing electronic medical records (which are subject to regular audit and verification) using a standardized data collection form. Collected variables included demographics (age, ethnicity, BMI, parity, breastfeeding history, smoking, and contraceptive use), clinical presentation (presence of mass, abscess, sinus tract, flares), imaging findings (mass size, margins, vascularity, calcifications, lymphadenopathy), histopathology (presence of necrosis, granulomas, organisms cultured), and treatment details (steroid dose and duration, methotrexate use and dose, surgical interventions).

### 2.4. Treatment Definitions and Outcomes

In this study, steroid-only therapy was defined as the use of oral corticosteroids (specifically prednisolone) without any additional immunomodulatory agents. Step-up therapy referred to the addition of methotrexate following initial corticosteroid treatment, typically due to inadequate clinical response, disease relapse, or the development of significant steroid-related side effects. Surgical intervention was considered when medical management failed and was defined as any of the following procedures performed to control disease: incision and drainage, excision biopsy, or saucerization.

The primary outcome of interest was the identification of clinical, imaging, and pathological factors predictive of the need for step-up therapy. Secondary outcomes included treatment success, defined as complete remission; disease relapse; treatment failure necessitating surgery; time to achieve remission; and time to surgery from the time of diagnosis

Complete remission was defined as the resolution of clinical signs and symptoms of IGM without the need for further medical or surgical intervention. Relapse was defined as the recurrence or flare of IGM symptoms following initial improvement during the course of treatment. Treatment failure referred to the persistence or progression of disease that ultimately required surgical management.

### 2.5. Statistical Analysis

Descriptive statistics were used to summarize patient characteristics and outcomes. Continuous variables were reported as medians (25th Percentile–75th Percentile). Categorical variables were summarized as frequencies and percentages.

To assess the association of the clinical characteristics, imaging findings, pathology and treatment options with the need for step-up therapy in idiopathic granulomatous mastitis, the univariate analyses using modified Poisson regression were performed. Relative risks (RRs) and 95% confidence intervals (CIs) were presented.

Modified Poisson regression was also carried out to compare those who were administered steroids to those who were administered methotrexate in experiencing complete remission, had surgery and did not have relapse, taking into account the change of treatment over the duration of the treatment period.

The univariate analyses on time to complete remission and time to surgery have been performed using Cox time-dependent regression to compare those who were administered steroids to those who were administered methotrexate. Multivariate analysis has also been further performed on time to complete remission adjusting for age, parity, presence of mass, largest mass diameter, vascularity, posterior shadowing, calcification, sinus tract presence and margin irregularity. Hazard ratios (HRs) and 95% CIs were presented.

The statistical evaluations were performed at a 2-sided test and 5% of level of significance. Analyses have been conducted using IBM SPSS version 29 and R software version 4.4.1.

A complete case analysis was performed as missing data were minimal (<5% for any variable). For sensitivity, variables with occasional missingness (e.g., imaging features) were examined for patterns of missingness (missing completely at random [MCAR], missing at random [MAR], or missing not at random [MNAR]). No imputation was applied as patterns were random and the proportion low.

### 2.6. Ethical Approval

This study and its conduct were approved by the National Health Group Domain Specific Review Board (DSRB Reference No: 2024-3264).

## 3. Results

### 3.1. Study Population and Characteristics

A total of 77 patients were screened between May 2022 and June 2024. Of these, 52 women met inclusion criteria, with 25 excluded due to alternative diagnoses (e.g., tuberculosis, foreign body reaction) or incomplete records (Figure 1). The median follow-up duration calculated from the date of diagnosis was 13.8 months (25th Percentile–75th Percentile: 7.1–23.2 months). The median age at diagnosis was 39 years (IQR: 32–42 years).

Among the 52 women, 30 (57.7%) were Chinese, 16 (30.8%) Malay, 3 (5.8%) Indian, and 3 (5.8%) of other ethnicities. Most patients were premenopausal (94.2%) and parous (67.3%), with 36.5% having breastfed recently. Oral contraceptive use at diagnosis was reported in 2 patients (3.8%), and 5 patients (9.6%) were smokers. The majority (78.8%) had a BMI > 23 kg/m^2^. A breast mass was the most common presenting feature, seen in 96.2% of cases, while 30.8% experienced at least one flare during follow-up. Representative clinical presentations are shown in Figure 2A,B. These include a breast abscess with overlying skin desquamation (Figure 2A) and a breast abscess that spontaneously drained through a fistula (Figure 2B).

### 3.2. Imaging and Pathological Findings

On ultrasound, 80.8% (*n* = 42) of the lesions were hypoechoic, 90.4% (*n* = 47) exhibited irregular margins, and 53.8% (*n* = 28) had increased internal vascularity. Other features included posterior shadowing (9.6%, *n* = 5), calcifications (9.6%, *n* = 5), sinus tracts (7.7%, *n* = 4), and lymphadenopathy (7.7%, *n* = 4).

The majority (82.7%) of breast tissue cultured were sterile. The most common organism cultured was *Corynebacterium kroppenstedtii*, found in 9 (17.3%) patients. Out of 52 patients, 37 (71.2%) initially received oral antibiotics for presumed breast infection. Of these, 19 (51.4%) did not improve and required steroid-only or step-up therapy after a median of 1.6 weeks.

### 3.3. Treatment Modalities

Of the 52 patients, 48 (92.3%) received corticosteroids as initial treatment. Among these, 22 (45.8%) were managed with steroid-only therapy, and 26 (54.2%) required step-up therapy with methotrexate after a median of 14 months (range not specified). Four patients (7.7%) declined medical treatment, citing concerns about steroid side effects. Two of these achieved spontaneous remission, while the other two showed partial improvement and remained under surveillance.

Among those treated, the median starting prednisolone dose was 20 mg/day (range 5–40 mg/day), with a median duration of 5 months in the steroid-only group. Further details on the steroid dosing schedule can be found in Table 1. The median starting methotrexate dose was 12.5 mg/week (range 2.5–15 mg/week). Surgery was performed in four patients (7.7%) due to treatment failure, consisting of three incision and drainage procedures and one saucerization. All surgical cases subsequently achieved remission. Details comparing treatment types against outcomes can be found in Table 2.

Accordingly, there were a few reasons why the 26 patients required step-up therapy. The first reason was the side effects from steroid-only therapy such as moon facies in 3, weight gain in 3, worsening of diabetes in 1, giddiness in 1, and relapse of psychosis in 1. The second reason was the progression of disease during the course of steroids, where there were new lumps in 6, larger existing lumps in 3, worsening breast pain in 3, new discharge in 2, perforation of lump in 1. The third reason was the lack of efficacy of steroids, described as persistent lumps in 4, and persistent fistulas in 1. The fourth reason was that some patients were unable to taper their dose of steroids without experiencing a recurrence of symptoms. This is described as an occurrence of flares in 7. No patients developed side-effects on methotrexate during the period of study.

### 3.4. Predictors of Step-Up Therapy

Oral contraceptive use at the time of diagnosis was significantly associated with an increased risk of requiring step-up therapy, with a RR of 1.92 (95% CI 1.45–2.53; *p* < 0.001). Similarly, smoking was associated with a higher likelihood of needing step-up treatment (RR 2.00, 95% CI 1.49–2.69; *p* < 0.001). The presence of flares during follow-up further increased the risk (RR 2.33, 95% CI 1.44–3.79; *p* = 0.002), as did the need for percutaneous aspiration (RR 2.10, 95% CI 1.53–2.88; *p* = 0.025). In contrast, other demographic, clinical, imaging, or pathological features, including age, body mass index (BMI), mass size, internal vascularity, and margin irregularity, were not statistically significantly associated with step-up therapy after adjustment in the regression model (Table 3).

### 3.5. Treatment Outcomes

There were 26 patients receiving step-up methotrexate therapy. Out of these 26 patients, 17 patients did not experience relapse before and after receiving methotrexate therapy while 9 patients experienced relapse before receiving methotrexate therapy. Patients who received methotrexate therapy had a higher chance of no relapse compared to patients who received steroids (RR: 1.23, 95% CI 1.12–1.36, *p* < 0.001). There was no statistically significant difference in experiencing complete remission (RR: 1.03, 95% CI 0.48–2.25, *p* = 0.993) and risk of surgery (RR: 0.63, 95% CI 0.07–5.35, *p* = 0.672) between those who received steroids and methotrexate.

### 3.6. Time-to-Event Outcomes

There was boundary statistically significant difference in time to complete remission between those who had methotrexate and steroid (Hazard Ratio: 0.33, 95% CI 0.11 to 1.01, *p* = 0.051). After adjusting for age, parity, presence of mass, largest mass diameter, vascularity, posterior shadowing, calcification, sinus tract presence and margin irregularity, those who had methotrexate had statistically significant longer time to complete remission compared to those who had steroid (Hazard Ratio: 0.09, 95% CI 0.02 to 0.46, *p* = 0.004). There was no statistically significant difference in time to surgery between those who received steroid and methotrexate (HR 3.63, 95% CI 0.20–64.67; *p* = 0.38).

## 4. Discussion

Approximately half of the patients in this study cohort required step-up therapy with methotrexate after initial corticosteroid treatment. Oral contraceptive use, smoking, the presence of flares, and the need for percutaneous aspiration emerged as significant independent predictors of the need for step-up therapy. These results support the rationale for earlier initiation of methotrexate in patients in high-risk subgroups. Patients who received methotrexate had a higher chance of no relapse compared to those receiving steroids although they had a longer time to achieve complete remission, likely reflecting delays in initiating immunomodulation.

IGM is a rare and challenging condition primarily affecting women of childbearing age. Consistent with prior literature, most of the patients in our cohort were premenopausal, parous, and had recent breastfeeding history, which are established risk factors for IGM [5,15,24]. Oral contraceptive use at diagnosis, smoking, presence of flares, and the need for percutaneous aspiration were identified as significant independent predictors of requiring step-up therapy. These associations are biologically plausible and align with existing evidence [1,4,16,25,26,27,28]. Hormonal influences of oral contraceptives may contribute to immune modulation and higher rates of relapse [29,30]. Similarly, smoking is well-recognized as a risk factor for periductal mastitis and recurrent abscesses and may exacerbate IGM through its effects on immune dysregulation and local tissue damage.

The presence of flares and the requirement for aspiration likely reflect more severe or refractory disease at presentation, necessitating escalation of therapy. Our study found that methotrexate, a repurposed immunomodulator, confers superior relapse prevention compared to corticosteroids alone in IGM. This is an incremental advance for the growing body of evidence on IGM treatment and management [18,27,30,31]. Patients who received methotrexate experienced higher chance of no relapse compared to patients who received steroid (RR: 1.23, 95% CI 1.12–1.36, *p* < 0.001). This finding reinforces the utility of methotrexate as a steroid-sparing agent and a valuable option in managing steroid-resistant or intolerant cases. However, it was observed that patients who received methotrexate was associated with a longer time to achieve complete remission (Figure 3), with an adjusted hazard ratio of 0.09 (95% CI 0.02–0.46; *p* = 0.004) compared to patients who received steroids. This may reflect delays in initiating methotrexate, as patients typically received it only after prolonged corticosteroid use had proven insufficient or poorly tolerated. Such delays could prolong the overall disease course and contribute to cumulative morbidity. Potential adverse events of methotrexate include hepatotoxicity, cytopenia and teratogenicity. While these complications were not observed during the study period, careful patient selection, monitoring of liver function and blood counts, and counselling for contraception remain essential if methotrexate were to be considered as first-line therapy.

Importantly, earlier introduction of methotrexate in high-risk patients could represent a patient-centric, risk-stratified approach, reducing steroid burden and potentially averting surgery. These findings underscore the potential value of early identification of patients at high risk of treatment failure and consideration of a top-down approach, where immunomodulation is introduced earlier in the course of treatment rather than as salvage therapy. Importantly, our findings suggest that a more proactive, risk-stratified approach to IGM treatment could improve patient outcomes. For patients with identifiable risk factors for step-up therapy, earlier initiation of methotrexate may help prevent repeated flares, minimize cumulative steroid exposure, and reduce the need for surgical intervention. While this strategy may be beneficial, prospective studies with larger cohorts are required before any firm recommendations can be made. Although our study did not find statistical significant differences in surgical rates between those who received methotrexate and steroids (RR: 0.63; 95% CI 0.07–5.35; *p* = 0.672), the result that showed those with methotrexate had higher chance of no relapse (RR:1.23; 95% CI 1.12–1.36; *p* < 0.001) nonetheless support its role in achieving more durable disease control (Figure 4).

There are several arising implications for future research. While methotrexate appears to be an effective adjunct, its potential as a first-line agent, either alone or in combination with corticosteroids, remains to be clarified. Given its relatively favorable side effect profile, apart from its teratogenicity, methotrexate could represent an attractive option for carefully selected patients, particularly those with contraindications to long-term corticosteroid use. Prospective studies and randomized controlled trials comparing steroid monotherapy, methotrexate monotherapy, and combination therapy are needed to confirm the optimal treatment algorithm for IGM.

### Study Limitations

Nevertheless, this study has several limitations that should be acknowledged. First, the retrospective design introduces the potential for selection bias, information bias, and residual confounding, as treatment decisions and follow-up were not standardized but instead reflected real-world clinical practice. Second, the relatively small sample size, although reasonable for a single-center study of this rare condition, limits the precision of estimates and the ability to conduct more granular subgroup analyses. Hence, the study is primarily powered to detect moderate to large effect sizes in predictors of step-up therapy. Third, the absence of patients treated with methotrexate as first-line monotherapy precluded a direct comparison of methotrexate monotherapy, steroid monotherapy, and combination therapy. As a result, the potential role of methotrexate as an initial treatment option could not be evaluated. Another limitation is the variability in the timing and indications for initiating methotrexate, which reflected clinician judgment rather than a protocolized approach. This may have contributed to the observed longer time to remission in the methotrexate group, as methotrexate was often introduced only after prolonged corticosteroid use had proven insufficient or poorly tolerated. Additionally, although the follow-up duration was reasonable for most patients, it may still be insufficient to capture late relapses, long-term adverse outcomes, or sustained remission. Finally, as this was a single-center study, the findings may not be generalizable to other settings with different patient populations, clinical practices, or healthcare systems. Future multicenter prospective studies, ideally with standardized treatment protocols and longer follow-up, are needed to validate these findings and provide stronger evidence to guide clinical practice.

## 5. Conclusions

In summary, while corticosteroids remain the cornerstone of initial treatment, our findings show that methotrexate is an effective therapeutic adjunct for patients with refractory IGM disease or those experiencing significant side effects or steroid toxicity. The lower relapse rates and more durable disease control observed are encouraging, and these results support a move toward more personalized, risk-stratified management of IGM, where earlier initiation of immunomodulation could improve outcomes and minimize unnecessary corticosteroid exposure. Future prospective, multicenter studies and randomized controlled trials are essential to validate these predictors, define the optimal timing and confirm the role of methotrexate, and establish evidence-based treatment algorithms for this challenging condition.

## Figures and Tables

**Figure 1 jcm-14-07157-f001:**
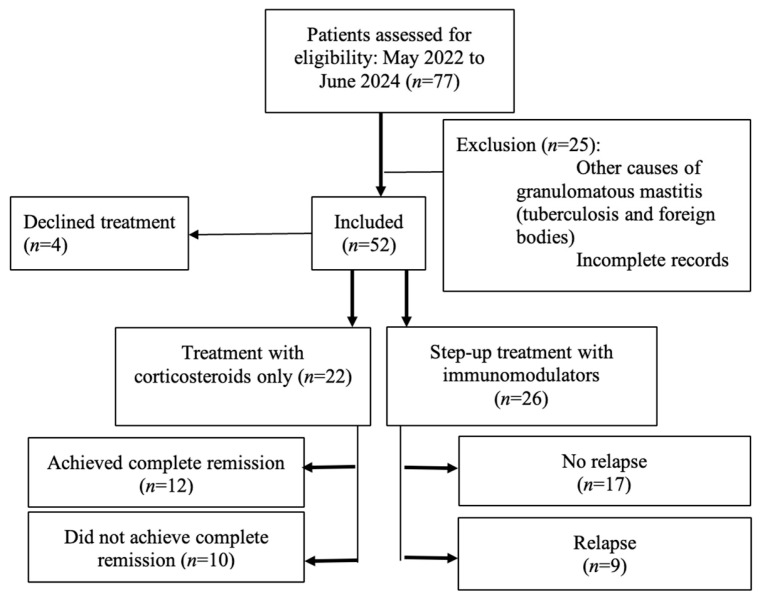
STROBE flow diagram showing patient inclusion, exclusions, and treatment. allocation. A total of 77 patients were screened between May 2022 and June 2024. Of these, 52 women met inclusion criteria, with 25 excluded (23 excluded due to alternative diagnoses (e.g., tuberculosis, foreign body reaction) and 2 excluded due to incomplete records). Of the 52 women who were included, 4 declined treatment, 22 received treatment with corticosteroids only, and 26 received step-up treatment with initial corticosteroids and immunomodulators. Of the 22 women who received treatment with corticosteroids only, 12 achieved complete remission, while 10 did not receive complete remission. In comparison, of the 26 women who received step-up treatment, 17 did not experience relapse before and after receiving methotrexate therapy, while 9 patients experienced relapse before receiving methotrexate therapy.

**Figure 2 jcm-14-07157-f002:**
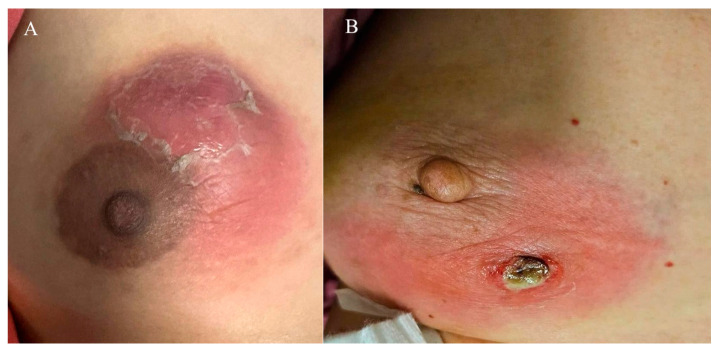
(**A**) Breast abscess with overlying desquamation of skin. (**B**) Breast abscess that has spontaneously drained through a fistula.

**Figure 3 jcm-14-07157-f003:**
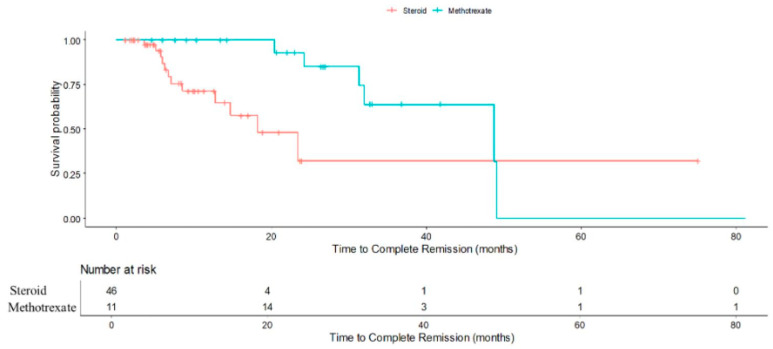
Kaplan-Meier curve demonstrating cumulative probability of complete remission in steroid-only and methotrexate (after initial steroids) patients.

**Figure 4 jcm-14-07157-f004:**
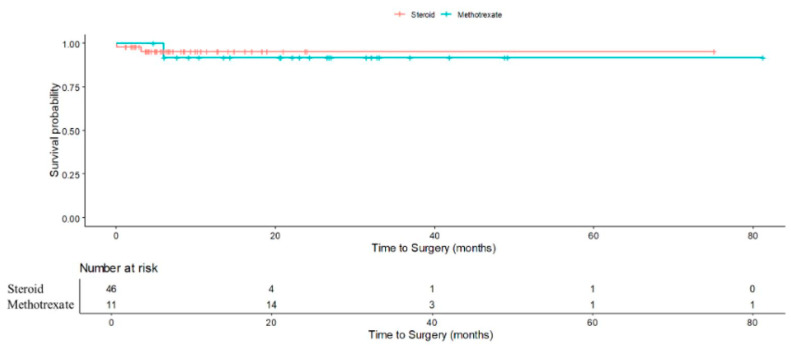
Kaplan-Meier curve demonstrating cumulative probability of surgery in steroid-only vs. methotrexate (after initial steroids) patients.

**Table 1 jcm-14-07157-t001:** Treatment regimen and outcomes.

	Steroids Only *n* (%)	Methotrexate (After Initial Steroids) *n* (%)
Total no. of patients	22 (100)	26 (100)
Median starting dose	20 mg/day	12.5 mg/week
No. of patients with complete remission	15 (68.2)	8 (30.7)
No. of patients with relapse	2 (9.1)	9 (24.6)
No. of patients with treatment failure	2 (9.1)	2 (7.7)

**Table 2 jcm-14-07157-t002:** Comparing treatment types against treatment outcomes.

	Steroids Only OR (95%CI), *p*-Value (*n* = 22)	Stepup Therapy OR (95%CI), *p*-Value (*n =* 26)
Complete remission No complete remission	3.7 (1.2–12.6), 0.026 1.0	0.27 (0.080–0.83), 0.026 1.0
Relapse No relapse	0.56 (0.020–0.64), 0.021 1.0	7.0 (1.6–50), 0.021 1.0
Treatment failure (surgery) No surgery	0.087 (0.10–8.2), 0.94 1.0	0.080 (0.12–9.7), 0.94 1.0

**Table 3 jcm-14-07157-t003:** Clinical characteristics, imaging findings, pathology and treatment options for patients on steroid-only therapy and step-up therapy. Log-binomial regression analysis of predictors associated with the need for step-up therapy in idiopathic granulomatous mastitis. The covariates included in the multivariate regression analyses: age, parity, presence of mass, largest mass diameter, vascularity, posterior shadowing, calcification, sinus tract presence, and margin irregularity.

Characteristic of Patients with IGM		Total Patients, *n* (%)	No Treatment, *n* (%)	Steroid-Only, *n* (%)	Step-Up, *n* (%)	RR for Step-Up with Respect to Steroid-Only (95% CI)	*p*-Value
Clinical Characteristics	Age at Diagnosis (years) >40 ≤40	18 (34.6) 34 (65.4)	0 (0) 4 (11.8)	10 (55.6) 12 (35.3)	8 (44.4) 18 (52.9)	0.74 (0.41 to 1.34) ref	0.375
	Race Chinese Malay Indian Others	30 (57.7) 16 (30.8) 3 (5.8) 3 (5.8)	3 (10.0) 0 (0) 0 (0) 1 (33.3)	15 (50.0) 5 (31.3) 1 (33.3) 1 (33.3)	12 (40.0) 11 (68.8) 2 (66.7) 1 (33.3)	ref 1.55 (0.91 to 2.64) 1.50 (0.61 to 3.71) 1.13 (0.26 to 4.79)	0.440
	BMI (kg/m^2^) >23 ≤23	41 (78.8) 11 (21.2)	3 (7.3) 1 (9.1)	18 (43.9) 4 (36.4)	20 (48.8) 6 (54.5)	0.88 (0.49 to 1.58) ref	0.735
	Menopausal Status Pre-menopausal Post-menopausal	49 (94.2) 3 (5.8)	3 (6.1) 1 (33.3)	21 (42.9) 1 (33.3)	25 (51.0) 1 (33.3)	1.09 (0.27 to 4.46) ref	1.000
	Previous Pregnancy Yes No	35 (67.3) 17 (32.7)	3 (8.6) 1 (5.9)	16 (45.7) 6 (35.3)	16 (45.7) 10 (58.8)	0.8 (0.48 to 1.34) ref	0.542
	Breastfeeding at diagnosis Yes No	19 (36.5) 33 (63.5)	1 (5.3) 3 (9.1)	7 (36.8) 15 (45.5)	11 (57.9) 15 (45.5)	1.22 (0.73 to 2.04) ref	0.555
	Oral Contraceptive Use at Diagnosis Yes No	2 (3.8) 50 (96.2)	0 (0) 4 (8.0)	0 (0) 22 (44.0)	2 (100) 24 (48.0)	1.92 (1.45 to 2.53) ref	<0.001
	Smoking Status Yes No	5 (9.6) 47 (90.4)	1 (20.0) 3 (6.4)	0 (0) 22 (46.8)	4 (80.0) 22 (46.8)	2 (1.49 to 2.69) ref	<0.001
	Presenting Complaint (Lump) Yes No	44 (84.6) 8 (15.4)	4 (9.1) 0 (0)	18 (40.9) 4 (50.0)	22 (50.0) 4 (50.0)	1.10 (0.52 to 2.32) ref	1.000
	Presence of Flares Yes No	16 (30.8) 36 (69.2)	0 (0) 4 (11.1)	2 (12.5) 20 (55.6)	14 (87.5) 12 (33.3)	2.33 (1.44 to 3.79) ref	0.002
Imaging Characteristics	Presence of Mass Yes No	50 (96.2) 2 (3.8)	4 (8.0) 0 (0)	21 (42.0) 1 (50.0)	25 (50.0) 1 (50.0)	1.09 (0.27 to 4.46) ref	1.000
	Hypoechoic Mass Yes No	42 (80.8) 10 (19.2)	4 (9.5) 0 (0)	19 (45.2) 3 (30.0)	19 (45.2) 7 (70.0)	0.71 (0.43 to 1.20) ref	0.307
	Largest Diameter of Mass (cm) >4 ≤4	28 (53.8) 24 (46.2)	1 (3.6) 3 (12.5)	10 (35.7) 12 (50.0)	17 (60.7) 9 (37.5)	1.47 (0.83 to 2.60) ref	0.244
	Irregular Margins Yes No	47 (90.4) 5 (9.6)	4 (8.5) 0 (0)	19 (40.4) 3 (60.0)	24 (51.1) 2 (40.0)	1.40 (0.46 to 4.22) ref	0.649
	Increased Internal Vascularity Yes No	28 (53.8) 24 (46.2)	2 (7.1) 2 (8.3)	9 (32.1) 13 (54.2)	17 (60.7) 9 (37.5)	1.60 (0.90 to 2.84) ref	0.146
	Posterior Shadowing Yes No	5 (9.6) 47 (90.4)	0 (0) 4 (8.5)	2 (40.0) 20 (42.6)	3 (60.0) 23 (48.9)	1.12 (0.52 to 2.42) ref	1.000
	Calcifications Yes No	5 (9.6) 47 (90.4)	0 (0) 4 (8.5)	2 (40.0) 20 (42.6)	3 (60.0) 23 (48.9)	1.12 (0.52 to 2.42) ref	1.000
	Subcutaneous Oedema Yes No	4 (7.7) 48 (92.3)	0 (0) 4 (8.3)	2 (50.0) 20 (41.7)	2 (50.0) 24 (50.0)	0.92 (0.33 to 2.53) ref	1.000
	Sinus Tract Yes No	4 (7.7) 48 (92.3)	1 (25.0) 3 (6.3)	2 (50.0) 20 (41.7)	1 (25.0) 25 (52.1)	0.60 (0.12 to 3.04) ref	0.587
	Lymphadenopathy Yes No	4 (7.7) 48 (92.3)	0 (0) 4 (8.3)	2 (50.0) 20 (41.7)	2 (50.0) 24 (50.0)	0.92 (0.33 to 2.53) ref	1.000
Pathological Characteristics	Presence of Necrosis Yes No	4 (7.7) 48 (92.3)	0 (0) 4 (8.3)	2 (50.0) 20 (41.7)	2 (50.0) 24 (50.0)	0.92 (0.33 to 2.53) ref	1.000
	Presence of Granuloma Yes No	22 (42.3) 30 (57.7)	0 (0) 4 (13.3)	9 (40.9) 13 (43.3)	13 (59.1) 13 (43.3)	1.18 (0.70 to 1.98) ref	0.573
	Abscess Formation Yes No	14 (26.9) 38 (73.1)	0 (0) 4 (10.5)	5 (35.7) 17 (44.7)	9 (64.3) 17 (44.7)	1.29 (0.77 to 2.15) ref	0.526
Other Treatment Options	Microorganisms cultured Yes No	9 (17.3) 43 (82.7)	0 (0) 4 (9.3)	5 (55.6) 17 (39.5)	4 (44.4) 22 (51.2)	0.79 (0.36 to 1.72) ref	0.713
	Percutaneous Aspiration Yes No	6 (11.5) 46 (88.5)	0 (0) 4 (8.7)	0 (0) 22 (47.8)	6 (100) 20 (43.5)	2.10 (1.53 to 2.88) ref	0.025

## Data Availability

The datasets generated and/or analyzed during the current study are not publicly available due to ethical reasons but are available from the corresponding author on reasonable request. All shared data will be provided in de-identified form to ensure participant privacy.

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
