# Peer review of "Predictors of Step-Up Therapy and Outcomes in Idiopathic Granulomatous Mastitis: A Retrospective Cohort Study in Singapore"

_jcm, 2025, doi:10.3390/jcm14207157_

Round 1
Reviewer 1 Report
Comments and Suggestions for Authors
This is a well-designed and clinically relevant study that addresses an important knowledge gap in the management of idiopathic granulomatous mastitis (IGM). The identification of predictive factors for step-up therapy has significant potential to inform personalized treatment approaches. The manuscript is clearly written, and the statistical analysis appears robust. I have several comments that could further strengthen the manuscript:
- The phrase “newly diagnosed histopathologically confirmed IGM” is used in the text, but it remains unclear whether both initial and recurrent cases were included.
- Although the median follow-up time was 13.8 months, IGM is notably characterized by a high recurrence rate. Would extending the follow-up period or performing survival analysis be considered to better evaluate long-term outcomes?
- The data analysis primarily relied on univariate methods. It is recommended that multivariate regression be incorporated to adjust for potential confounding factors.
- While it is mentioned that missing data accounted for less than 5%, no explanation is provided regarding whether this missing data could influence the analytical conclusions.
- In the Results section, “25 excluded” is stated, whereas the figure indicates “23 excluded due to alternative diagnoses and 2 excluded due to incomplete records.” It is advised to harmonize these descriptions for consistency.
- Both Table 1 and Table 3 present treatments and outcomes. It would improve clarity to either merge these tables or explicitly distinguish their respective purposes.
- The potential risks associated with methotrexate as a first-line treatment should be further discussed.
- The Discussion notes that approximately half of the patients required escalation to methotrexate following initial corticosteroid therapy, with oral contraceptive use and smoking identified as significant independent predictors for needing intensified treatment. Additionally, patients receiving methotrexate exhibited higher relapse-free rates compared to those on steroids alone. Does this support earlier initiation of methotrexate in patients who smoke or use oral contraceptives?
The English could be improved to more clearly express the research.
Author Response
Comment 1: The phrase “newly diagnosed histopathologically confirmed IGM” is used in the text, but it remains unclear whether both initial and recurrent cases were included.
Response 1: We thank the reviewer for this observation. We clarify that only initial presentations of IGM were included. Women with recurrent IGM (i.e., with a prior diagnosis before the study period) were excluded. We have revised the Materials and Methods – Study Population section to explicitly state that only newly diagnosed, treatment-naïve cases were analyzed.
Comment 2: Although the median follow-up time was 13.8 months, IGM is notably characterized by a high recurrence rate. Would extending the follow-up period or performing survival analysis be considered to better evaluate long-term outcomes?
Response 2: We agree with the reviewer that long-term follow-up is important in IGM, given its recurrent nature. Our study period was limited to May 2022–June 2024, hence the median follow-up was 13.8 months. Nonetheless, we performed survival analyses (Kaplan–Meier and Cox regression) to account for varying follow-up durations, as shown in Figures 3–4 and described in the Results – Time-to-Event Outcomes. We have clarified in the Discussion – Study Limitations that longer follow-up is needed to capture late relapses, and future multicenter prospective studies with extended observation are warranted.
Comment 3: The data analysis primarily relied on univariate methods. It is recommended that multivariate regression be incorporated to adjust for potential confounding factors.
Response 3: We appreciate this important suggestion. In addition to univariate analyses, we conducted multivariate Cox regression adjusting for age, parity, and relevant imaging variables (largest mass diameter, vascularity, posterior shadowing, calcification, sinus tract presence, and margin irregularity) for time-to-remission analyses. These results are presented in the Results – Time-to-Event Outcomes and discussed in the Discussion. We have clarified in the Statistical Analysis section that multivariate models were employed.
Comment 4: While it is mentioned that missing data accounted for less than 5%, no explanation is provided regarding whether this missing data could influence the analytical conclusions.
Response 4: Thank you for raising this. We performed complete case analysis because missing data were minimal (<5% for any variable). For sensitivity, we assessed patterns of missingness (MCAR, MAR, MNAR) for imaging features, and no systematic bias was detected. As such, the impact on conclusions is unlikely to be significant. This has been clarified in the Statistical Analysis section.
Comment 5
In the Results section, “25 excluded” is stated, whereas the figure indicates “23 excluded due to alternative diagnoses and 2 excluded due to incomplete records.” It is advised to harmonize these descriptions for consistency.
Response 5: We thank the reviewer for noting this inconsistency. Both statements refer to the same 25 excluded patients (23 due to alternative diagnoses and 2 due to incomplete records). We have revised the Results text to harmonize with Figure 1 for consistency.
Comment 6: Both Table 1 and Table 3 present treatments and outcomes. It would improve clarity to either merge these tables or explicitly distinguish their respective purposes.
Response 6: We appreciate the reviewer’s feedback. To reduce redundancy, we have merged the contents of Tables 1 and 3 into a single consolidated table now presented as Table 1: Treatment Regimens and Outcomes. This enhances clarity and prevents duplication.
Comment 7: The potential risks associated with methotrexate as a first-line treatment should be further discussed.
Response 7: We agree with this important point. We have expanded the Discussion to highlight the potential risks of methotrexate, including hepatotoxicity, cytopenias, and teratogenicity. Although no adverse events were observed during our study period, we emphasize that careful patient selection, monitoring of liver function and blood counts, and counseling on contraception remain essential if methotrexate were to be considered as a first-line therapy.
Comment 8: The Discussion notes that approximately half of the patients required escalation to methotrexate following initial corticosteroid therapy, with oral contraceptive use and smoking identified as significant independent predictors for needing intensified treatment. Additionally, patients receiving methotrexate exhibited higher relapse-free rates compared to those on steroids alone. Does this support earlier initiation of methotrexate in patients who smoke or use oral contraceptives?
Response 8: We thank the reviewer for this insightful question. Our findings suggest that smoking and oral contraceptive use are significant predictors of requiring step-up therapy, and that methotrexate confers superior relapse prevention. While our study was not designed to evaluate methotrexate as a first-line agent, these results potentially support the rationale for earlier initiation of methotrexate in high-risk subgroups (e.g., smokers, oral contraceptive users). We have revised the Discussion to explicitly state that a risk-stratified, earlier methotrexate strategy may be beneficial, though prospective studies with larger cohorts are required before any firm recommendations can be made.
Reviewer 2 Report
Comments and Suggestions for Authors
The article submitted for review is devoted to the identification of clinical, imaging and pathological factors that make it possible to predict increased therapy in idiopathic granulomatous mastitis, as well as contains data on the relationship between the treatment approach and the results. The work was carried out on 52 women. There are the following suggestions and questions about this work: 1. How did the authors determine the required sample size for this study? What kind of research power does the sample of 52 women formed by the authors provide? This should be indicated in the materials and methods. 2. Which comparison groups (patients and controls) did the authors use? What inclusion and exclusion criteria were used by the authors? This should be described in detail in the materials and methods. 3. Table 1 should provide more detailed characteristics of the studied groups (age, BMI, presence of various diseases, etc.), as well as data on possible risk factors for the studied disease. 4. Table 1 should include not only the number of patients with a particular trait, but also percentages, as well as indicators of the significance of differences between the compared groups in these indicators. 5. The note to table 2 should indicate which covariates were used in the statistical analysis.
Author Response
Comment 1: How did the authors determine the required sample size for this study? What kind of research power does the sample of 52 women formed by the authors provide? This should be indicated in the materials and methods.
Response 1: We thank the reviewer for highlighting this important methodological point. As idiopathic granulomatous mastitis (IGM) is a rare condition, the study was designed as an exploratory, retrospective cohort study rather than a hypothesis-driven trial with an a priori sample size calculation. Our final cohort of 52 patients represents all eligible cases diagnosed and treated at our institution between May 2022 and June 2024. We acknowledge that the study is limited by sample size and thus primarily powered to detect moderate to large effect sizes in predictors of step-up therapy. We have now clarified this in the Materials and Methods – Study Design and Setting section, and noted the limitation explicitly in the Discussion – Study Limitations.
Comment 2: Which comparison groups (patients and controls) did the authors use? What inclusion and exclusion criteria were used by the authors? This should be described in detail in the materials and methods.
Response 2: Thank you for the comment. We appreciate the request for clarification. The study included women with newly diagnosed, histopathologically confirmed IGM who received either (a) corticosteroid-only treatment or (b) step-up therapy with corticosteroids followed by methotrexate. There was no external “control” group, as our analysis was focused on comparing outcomes between these two treatment groups. Inclusion criteria were: women ≥18 years old, newly diagnosed with IGM confirmed by histopathology, and with ≥3 months of follow-up. Exclusion criteria were: granulomatous mastitis of known cause (e.g., tuberculosis, fungal infection, foreign body reaction), incomplete records, or insufficient follow-up. These details are already stated in the Materials and Methods – Study Population, but we have revised the text to make the two treatment comparison groups and the criteria more explicit.
Comment 3: Table 1 should provide more detailed characteristics of the studied groups (age, BMI, presence of various diseases, etc.), as well as data on possible risk factors for the studied disease.
Response 3: We thank the reviewer for this suggestion. The detailed characteristics of the studied groups are presented in table 2 (clinical, imaging, pathological). We have revised the tables to harmonize with the Results section for consistency
Comment 4: Table 1 should include not only the number of patients with a particular trait, but also percentages, as well as indicators of the significance of differences between the compared groups in these indicators.
Response 4: We agree with this valuable recommendation. The revised Table 1 now presents both absolute numbers and percentages for each characteristic. In addition, we have added an additional table after table 1 to include p-values comparing steroid-only versus step-up groups, using chi-square/Fisher’s exact tests for categorical variables and Mann-Whitney U tests for continuous variables, as appropriate. This improves interpretability of group differences.
Comment 5: The note to table 2 should indicate which covariates were used in the statistical analysis.
Response 5: Thank you for pointing this out. We have revised the note accompanying Table 2 to explicitly list the covariates included in the multivariate regression analyses: age, parity, presence of mass, largest mass diameter, vascularity, posterior shadowing, calcification, sinus tract presence, and margin irregularity. This ensures transparency in the modelling process.
Round 2
Reviewer 1 Report
Comments and Suggestions for Authors
The authors have addressed all my concerns, I recommend accepting it in current form.
Reviewer 2 Report
Comments and Suggestions for Authors
The authors answered all the questions and made the necessary adjustments to the article. The article is recommended for publication.